# Self-Reported Antimicrobial Stewardship Practices in Primary Care Using the TARGET Antibiotics Self-Assessment Tool

**DOI:** 10.3390/antibiotics9050253

**Published:** 2020-05-14

**Authors:** Leah Ffion Jones, Neville Quinton Verlander, Donna Marie Lecky, Sabeen Altaf, Dirk Pilat, Cliodna McNulty

**Affiliations:** 1Primary Care and Interventions Unit, Public Health England, Gloucester GL1 1DQ, UK; leah.jones@phe.gov.uk (L.F.J.); Donna.Lecky@phe.gov.uk (D.M.L.); 2Statistics, Modelling and Economics Department, Public Health England, London NW9 5EQ, UK; neville.verlander@phe.gov.uk; 3Royal College of General Practitioners, London NW1 2FB, UK; sabeen.altaf@rcgp.org.uk (S.A.); dirk.pilat@nhs.net (D.P.)

**Keywords:** TARGET, antimicrobial resistance, antimicrobial stewardship, primary care, self-assessment, general practice

## Abstract

The self-assessment tool (SAT) is a 16-question self-report of antimicrobial stewardship practices in primary care, available in the TARGET (Treat Antibiotics Responsibly, Guidance Education and Tools) Antibiotics Toolkit. This study analysed responses to the SAT and compared them to previous SAT data (2014–2016). Data from June 2016 to September 2019 were anonymised and analysed using Microsoft Excel and STATA 15. Clinicians reported engaging in positive antimicrobial stewardship (AMS) practices including using antibiotic guidance to inform treatment decisions (98%, 98% 2014–2016), discussing antibiotic prescribing within the practice (73%, 67% 2014–2016), using patient-facing resources (94%, 71% 2014–2016), conducting antibiotic audits in the last two years (98%, 45% 2014–2016), keeping written records and action plans (81%, 62% 2014–2016), using back-up prescribing (99%, 94% 2014–2016) and using clinical coding (80%, 75% 2014–2016). Areas for improvement include developing strategies to avoid patients reconsulting to obtain antibiotics (45%, 33% 2014–2016), undertaking infection-related learning (37%, 29% 2014–2016), ensuring all temporary prescribers have access to antibiotic guidance (55%, 63% 2014–2016) and making patient information leaflets easily available during consultations (31%). The findings offer a unique insight into AMS in primary care over time. The SAT gives primary care clinicians and commissioners an opportunity to reflect on their AMS and learning.

## 1. Introduction

In 2012, Public Health England (PHE) and the Royal College of General Practitioners (RCGP) launched an antimicrobial stewardship (AMS) toolkit for primary care clinicians called the TARGET Antibiotics Toolkit. TARGET stands for Treat Antibiotics Responsibly; Guidance, Education and Tools and includes group and self-learning activities, patient information leaflets, diagnostic and management compliance audits, diagnostic flow charts and antibiotic guidance, patient-facing posters and videos, an AMS self-assessment tool (SAT) and online educational modules [1]. The SAT is a 16-item questionnaire made up of 12 multiple-choice questions covering current AMS practice and four open-ended reflective questions which allow respondents to reflect on their responses (See Appendix A for a screen shot of the SAT). After each question, there are information icons to reveal extra information on the topic area. The SAT is divided into three sections: what would be good practice now, what most practices should aim to do soon and what all antibiotic-aware practices should be doing.

Self-assessment is a common tool used to facilitate learning [2]. The process of self-assessment relies on the ability of an individual to identify discrepancies between their current and desired behaviour [3]. Reflection is a complementary tool which can be used to help individuals evaluate their skills and knowledge and how they can be developed [3,4]. Together, self-assessment and reflection are integral to developing competence as part of continued professional development [5].

The development of the SAT was detailed in a previous study [6]. The SAT is available as a standalone resource or as a non-compulsory precursor to the RCGP online eModule entitled “Antibiotic Resistance in Primary Care” (www.elearning.rcgp.org.uk). The eModule and SAT were originally made available to all who wished to access them; however, since February 2019 access to the eModule has been restricted to RCGP members only, but the standalone SAT on the TARGET website is still freely available. The SAT data were previously analysed from November 2014 to June 2016 from 1415 primary care clinicians from the UK [6]. The aim of this study is to present the most recent data collected from the SAT and provide a comparison with the previous findings to understand how AMS practice in primary care may have changed over time.

## 2. Results

In total, 2495 SATs were completed between July 2016 and September 2019 via the eModule Antibiotic Resistance in Primary Care (https://elearning.rcgp.org.uk/course/info.php?popup=0&id=167). Of those, 2111 (92%) were single responses, and 384 (15%) were second or third responses from the same 178 users. Of the 384 duplicates, responses were removed if they were given within a four-week period and the same responses were provided or if the duplicate response was incomplete. Finally, 262 duplicates were deemed to be legitimate. Therefore, 2373 responses were analysed.

Of the 2373 participants, 90.8% (2155) were based in a general practitioner (GP) practice, 4.3% (102) were based in ‘other’ locations, 2.4% (58) in out-of-hours (OOH) services, 2.4% (56) in hospital, one individual was based in a dental practice and one did not identify his/her profession.

Most respondents were GPs (1891, 79.7%), 11.3% (267) were nurses, 3.3% (78) pharmacists and 5.6% (134) reported ‘other’ as their job title; 70.9% (1683) were members of the RCGP, whereas 29.1% (690) were non-members.

Across the UK, 196 English Clinical Commissioning Groups (CCGs), 4 northern Irish health boards, 12 Scottish health boards and all 7 Welsh health boards were represented in the responses. The mean response per CCG or health board was 11 and ranged from 1 to 267.

### 2.1. What Would be Good Practice Now

Fifty-seven percent of respondents answered positively to all five questions in this section (Figure 1). Almost all respondents reported using national or local antibiotic guidance when considering how to treat common infections (98.8%, 2345). Seventy-three percent (1736) reported analysing and discussing antibiotic prescribing in their practice in comparison to local indicators at least once a year; nurses had higher odds of doing this compared to GPs (OR = 9.95, CI = 2.34–42.3).

Ninety-four percent (2238) of respondents reported using patient-focused strategies to highlight the importance of responsible antibiotic use, such as the use of videos and posters. Those who worked in hospitals, OOH and ‘other’ settings had lower odds of using such strategies than those in GP practices, although this was not significant for hospitals or OOH settings.

Ninety-eight percent (2329) of respondents reported being involved in a practice antibiotic audit in the last two years; of those, 57.4% (1362) had done so in the last year. Those working in ‘other’ settings and OOH had a higher propensity to be involved in an antibiotic audit in the last two years in comparison to those in GP practices, (coefficient = 0.94, CI = 0.22–1.66) (coefficient =1.35, CI = 0.45–2.24). Similarly, GPs had significantly lower propensity of having conducted an audit in the last two years compared to nurses and ‘other’ professionals (coefficient = 0.74, CI = 0.28–1.20), (coefficient = 1.90, CI = 1.49–3.77). Seventy-nine percent (1891) of respondents reported using clinical indications for antibiotics according to Read codes or SNOMED codes (which support detailed clinical encoding of patient phenomena such as signs and symptoms, diagnosis, treatment and patient demographics). Those working in OOHs (OR = 0.37, CI = 0.16–0.83), and other locations (OR = 0.28, CI = 0.13–0.62), had significantly lower odds of recording clinical indications for prescribed antibiotics compared to those working in GP practices.

These findings are illustrated by the following quotes taken from the reflective notes in question 4 and question 7:

Current practice: *“Use of delayed antibiotics, educating patients on resistance, discussing in clinical meeting, having self-care leaflets, audit use of antibiotics, educating trainees on antibiotics” GP, Sandwell and West Birmingham CCG.*

Current practice: *“We ensure our doctors are aware regarding current prescribing guidance—specifically antibiotic resistance—and avoid prescribing broad-range antibiotics. We audit prescribing for various conditions…We use posters and patient info leaflets regarding unnecessary and harmful antibiotic prescribing “GP, Ealing CCG.*

Clinical coding: *“Although I always read-code a diagnosis, I don’t necessarily record it correctly” GP, East Leicestershire and Rutland CCG.*

### 2.2. What Most Practices Should Aim to Do Soon

Just over half of the respondents (55.1%, 1308) reported that the latest antibiotic guidance was available to all temporary prescribers in their setting, although 23.3% (553) did not know. Nurses (OR = 2.39, CI = 1.32–4.34) and ‘others’ (OR = 4.27, CI = 1.49–12.2) had significantly higher odds of answering yes to this question compared to GPs.

Seventy-three percent (1739) of respondents reported using back-up antibiotics weekly, 11.9% (283) reported using them monthly, 14.4% (344) yearly, and less than 1% did not use or did not agree with back-up antibiotics. Hospital workers were significantly more likely than GPs to not use or not agree with using back-up antibiotics than to use them weekly (coefficient = 3.12, CI = 1.05–5.18). In contrast, nurses, pharmacists and ‘others’ were more likely to use back-up antibiotics.

A large proportion of respondents (68.9%, 1637) reported not having easily available leaflets to share in consultations, 29.4% (698) reported having hard copies and only 1.4% (33) reported having computer prompts. ‘Other’ settings, OOH and hospital settings were significantly more likely to have hard copies available than none at all, and ‘other’ professions had a significantly higher propensity to have hard copies available than none at all. For a visual summary of responses for this section, see Figure 2.

These findings are illustrated by the following quotes taken from the reflective notes in question 11:

Intention to use patient-facing leaflets: *“Don’t use patient leaflets at present, will start using ‘Treating your infection’ leaflet personally in consultations.” GP, Cwm Taf Health Board.*

Practice action planning: *“Our action plan at the last meeting was … to increase delayed prescribing and also give Public Health England “treating your infection” letters to patients and audit this activity.” GP, Sheffield CCG.*

### 2.3. What All Antibiotic-Aware Practices Should Be Doing

Most respondents (81.0%, 1923) reported keeping a written record and surgery action plan resulting from antibiotic audits. Those who worked in ‘other’ locations had significantly lower odds of keeping written records and a surgery action plan compared to those in a GP practice (OR = 0.43, CI = 0.26–0.73). Under half of the respondents (44.7%, 1061) reported having a strategy to avoid patients reconsulting with other clinicians to obtain antibiotics. Only 37% of respondents (875) reported undertaking an antibiotic-related prescribing clinical course. For a visual summary of this section, see Figure 3.

The final question, which asked respondents if their practice had a GP or nurse AMS lead, was added in April 2019, and only received 90 responses in total; of those, 53 answers were yes, and 37 were no.

These findings are illustrated by the following quotes taken from the reflective notes in question 16:

Participant attended local courses: *“Although I have not done an RCGP course, I have attended several meetings organised by the CCG where local Consultant Microbiologists have discussed management of RTI (respiratory tract infections) and UTI (urinary tract infections)” GP, Sandwell and West Birmingham CCG.*

Clinical course would be beneficial: *“I identify antibiotic prescribing as a huge component of my daily practice and feel that completing the suggested clinical courses online and also considering local strategy based on audit to avoid patient reconsulting with other clinicians will improve the surgery’s and my own prescribing outcomes” GP, Surrey Downs CCG.*

### 2.4. Comparison with 2014–2016 Data

Previous SAT data (November 2014 to June 2016) [6] compared with the data in this present study (June 2016–September 2019) can be seen in Table 1. Comparisons indicate that 98% of respondents reported using antibiotic guidance across both data sets. There was a small increase to 80% of respondents reporting using clinical indications and to 73% of respondents reporting analysing and discussing antibiotic prescribing in their practice.

Almost all respondents (99%) used back-up prescribing, compared to 94% in earlier data; the current data indicate that 73% were using back-up antibiotics weekly, and about 15% were only using them yearly. The current data indicate a 5% decrease, down to 55%, in reporting that the latest antibiotic guidance was made available to all temporary prescribers. There was a doubling of respondents reporting a practice antibiotic audit in the last two years, corresponding to 98%. There was a 23% and a 19% respective increase of respondents reporting using patient-focused strategies such as posters and videos (corresponding to 94%) and a written record and surgery action plan resulting from audits (corresponding to 81%).

Less than half (45%) of the respondents reported having a strategy to avoid patients reconsulting to obtain antibiotics, compared to the previously observed 33%. Thirty-seven percent of respondents reported undertaking antibiotic-related prescribing courses (previously, 29%).

Answers to question 10 about patient information leaflets cannot be compared to the previous data, as the question changed focus from ‘if leaflets were used’, to ‘if leaflets are easily available’. This does, however, highlight that over half of the respondents initially reported regularly using leaflets, but nearly 70% of them in the present study reported that leaflets were not easily available.

## 3. Discussion

### 3.1. Summary of the Findings

Many clinicians from across the UK reportedly engaged in positive AMS practice such as using antibiotic guidance to inform treatment decisions, discussing antibiotic prescribing within the practice, using patient-facing resources, conducting antibiotic-prescribing audits, keeping written records and action plans, using back-up prescribing and using clinical coding.

Areas with scope for improvement include developing strategies to help patients avoid reconsulting to obtain antibiotics, as fewer than half of the responded reported this. Nearly two-thirds of the respondents had not undertaken any antibiotic-related learning course in the last two years. Additionally, ensuring all temporary prescribers have access to antibiotic guidance could be improved, as nearly half of them reported that the guidance was not made available or they did not know about it. Similarly, having easily available patient information leaflets in consultations was reportedly low.

### 3.2. Comparison with Existing Literature

A comparison with previous data [6] shows that the use of antibiotic guidance, the use of clinical indications and discussions around antibiotic prescribing are consistently frequent. The high rates of reported guidance use could be due to local implementation efforts by CCG medicines management teams in England. According to a survey of CCGs, 83% of participants reported using national guidance to develop local prescribing guidelines, with most using multiple formats for promotion, including intranet platforms, websites and clinical systems [7]. Furthermore, the same survey found that all CCGs reportedly fed back local and/or national antimicrobial prescribing data to general practices, which may help facilitate antibiotic-prescribing discussions [8].

The present data indicate that 26.4% of respondents were using back-up prescribing only monthly or yearly, rather than weekly; therefore, interventions are needed if the NHSI wishes to maximise the use of back-up antibiotics. Nevertheless, three-quarters of the respondents reported using back-up prescribing weekly, as suggested by national guidance [9,10] and promoted locally by CCGs. In a CCG survey, 95% of participants reported promoting the use of back-up prescribing in their localities, [7] and national guidance recommends using a ‘no’ or ‘delayed’ strategy for self-limiting upper respiratory infections and mild urinary infections [11,12,13].

Only 55% of participants reported making the latest antibiotic guidance available to all temporary prescribers. This persistent low compliance with National Institute for Health and Care Excellence (NICE) guidance [10] highlights the need to facilitate access to the latest guidance for temporary prescribers, to ensure appropriate diagnosis and treatment in primary care. This is particularly important, as locums make up approximately 30% of fully licensed and registered GPs. Further research is needed to establish the preferred format of guidance for temporary prescribers.

The biggest improvement observed since the previous survey is in the use of antibiotic-prescribing audits. Further improvements can be seen in the use of patient-focused strategies and keeping a written record and surgery action plan as a result of audits. Audits combined with action planning have been shown to effectively improve practice [14]. In a survey of CCGs, 81% of participants reported undertaking AMS audits, 59% of which were reportedly carried out by practice staff. A further 89% of CCGs reported discussing the audit findings in GP practices [15]. Additionally, 94% of CCGs reported promoting AMS posters, and 58% reported promoting AMS videos [15]. Therefore, local action from CCGs may be driving the high use of audits, posters and videos and subsequent practice discussions around antibiotic prescribing.

Despite the improvements in the reported strategies to avoid patients reconsulting to obtain antibiotics and conducting further AMS-related learning, these activities were still relatively infrequent compared to other AMS activities. A possible explanation for the lack of engagement with antibiotic courses could be lack of awareness, as only 25% of CCGs in 2017 reported the use of compulsory antimicrobial stewardship/resistancelearning as part of their incentive schemes [15].

Despite CCGs promoting leaflet use [8] and some integrating them into GP systems, our findings show that GPs do not find leaflets easily accessible. This may reflect the increasing popularity of signposting to information portals and websites [16] or may suggest a lack of communication between CCGs and general practices on this matter, but more research is needed to understand the barriers to the implementation of patient leaflets.

### 3.3. Strengths and Limitations

The participants for this study were self-selected, as they participated in the TARGET online module ‘Antibiotic Resistance in Primary Care’. The respondents were therefore likely to have an interest in AMS and demonstrate more AMS behaviours compared to a random sample of primary care clinicians. Nevertheless, this study reports from a large sample of 2373 respondents, which accounts for over 5% of GPs in the UK.

A growing body of literature suggests that self-assessment is a flawed method for facilitating learning [17,18,19,20]. However, many studies that report flaws in self-assessment tend to use self-efficacy or perceptions of competence as the measure for self-assessing [2,20], whereas this present study uses self-reported AMS behaviour. One of the theories to explain why self-assessment is flawed is the ‘above-average effect’, whereby individuals assume that they are above average compared to others and therefore tend to assess themselves more favourably [21]. Nevertheless, self-reported behaviour too can be an unreliable objective measure of competence [22]. Therefore, the results from this study should be accepted with caution; however, the large negative responses for questions around antibiotic learning courses, patient information leaflets and strategies to avoid patients reconsulting do provide some reassurance that respondents were sincere.

The nature of the SAT is that it is designed as a learning tool for primary care clinicians, therefore any demographic questions superfluous to the learning experience are not included. For example, it would have been useful for research and data collection purposes to have probing closed questions around the nature of antibiotic audits and action plans, but this information was only collected in the reflective comments, which provide some insight, though not all respondents completed them or provided sufficient details.

### 3.4. Implications for Research and Practice

All participants for this study were UK-based; therefore, it is inappropriate to draw implications for AMS practices in other countries which have different health services and cultures influencing clinician behaviours. Other countries may wish to implement a similar AMS benchmarking tool which can simultaneously act as a learning tool for clinicians and an electronic method of data collection to inform intervention development. Whilst there are limitations associated with the data, the fact that the SAT is first and foremost learning tool for clinicians likely facilitates its widespread adoption. In future, for research purposes such as this, the SAT may benefit from the undergoing evaluation of its validity and reliability to ensure that the questions are measuring what it is hoped to measure [23].

The NICE advocates the use of a no or back-up prescribing for many common respiratory tract infections, and therefore primary care clinicians could increase their back-up antibiotic prescribing for these infections [9,10]. Some clinicians may be reluctant to use back-up prescribing strategies due to beliefs that they can cause further inconveniences to the patient and the clinician if a reconsultation is needed, concern that patients may find the instructions confusing, or concern that back-up prescriptions convey a contradictory message to the patients [24]. Resources to encourage the use of delayed prescribing can be found on the TARGET website, including a dedicated webinar and patient information leaflets [1]. The infection-related educational courses [25] should also receive greater promotion, and commissioners should also consider local shared-infection-related learning activities or workshops through Primary Care networks to increase the numbers of clinicians reporting recent learning around AMS.

Practice managers, AMS leads or practice clinical leads need to ensure that temporary prescribers have access to the latest diagnostic and treatment guidance when visiting their practice, make patient information leaflets readily available to ensure their use and develop a practice-wide strategy to avoid patients reconsulting to obtain antibiotics.

Further quantitative research using a large randomised sample is needed to corroborate and strengthen these findings, alongside qualitative research to understand the barriers to and facilitators of these behaviours in order to develop interventions to facilitate AMS. Further studies should directly measure the use of back-up antibiotics, posters and leaflets and the effect of some of these AMS activities on outcomes, such as antibiotic use.

## 4. Materials and Methods

Participants starting the RCGP eLearning module ‘Antibiotic resistance in primary care’ are required to complete the SAT digitally on the RCGP website prior to beginning the module. The RCGP eLearning team collated and anonymised the data from June 2016 to September 2019 and provided the data to the research team in a Microsoft Excel document.

Data were cleaned to remove test entries, half attempts and subsequent attempts within short time periods. A descriptive analysis was conducted in Microsoft Excel in order to compare the current data with previous data; mixed effects statistical regressions with Clinical Care commissioning Group (CCG) as upper level and respondent as lower level random effects, or just the CCG as random effect, as appropriate, and profession and workplace as fixed effects were conducted in Stata version 15. For the regression analyses, for the questions where the outcome was binary, logistic regression was used, whereas multinomial regression was used for those questions with more than two choices. Statistical significance was ascertained by means of the likelihood ratio test, with 5% taken as the significance level. Odds ratios (ORs) for the binary response questions and coefficients (coeffs) otherwise, together with their 95% confidence intervals (CIs), were obtained and are quoted in the results; full tables can be found in Appendix B. Due to the low response for question 15, this was not included in the statistical analysis.

Ethical approval was not required, as this was a service evaluation and all responses were anonymised by the RCGP by removing email addresses and any identifiable information.

## 5. Conclusions

Lack of a randomised sample limits the data; however, the findings offer a unique insight into the self-reported AMS behaviours in primary care and suggestions for ways to optimize AMS. It is reassuring to see that in the last three years, attention to AMS activities has not waned and, in many areas, has increased.

All practices should ensure that temporary prescribers have access to patient information leaflets and the latest diagnostic and treatment guidance and should develop a practice-wide strategy to avoid patients reconsulting to obtain antibiotics. Primary care clinicians should be issuing back-up antibiotic prescriptions more often and attend antibiotic-related prescribing courses. The SAT could be used by clinicians and CCGs to assess AMS in their practice or area. The SAT could be adapted to be used in other clinical settings and countries.

## Figures and Tables

**Figure 1 antibiotics-09-00253-f001:**
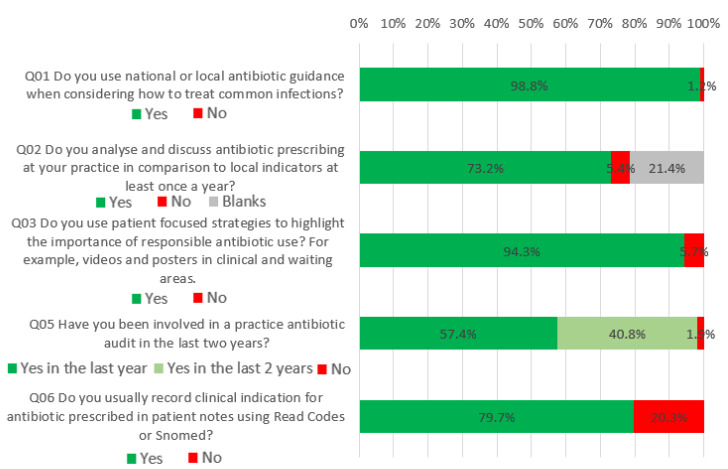
What Would be Good Practice Now.

**Figure 2 antibiotics-09-00253-f002:**
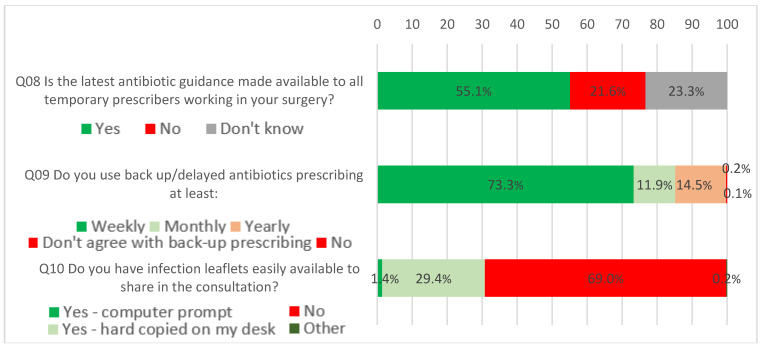
What Most Practices Should Aim to do Soon.

**Figure 3 antibiotics-09-00253-f003:**
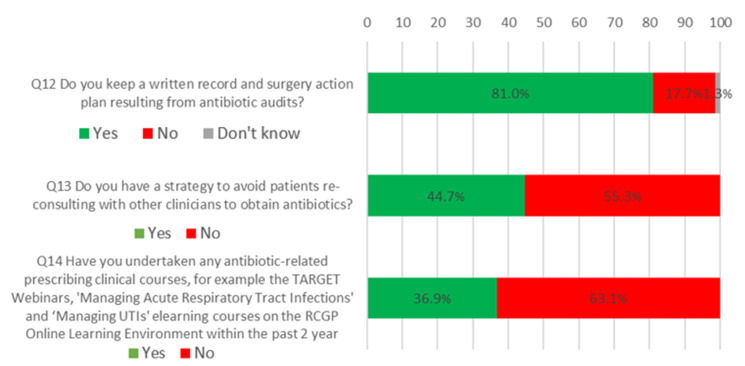
What all antibiotic-aware practices should be doing.

**Table 1 antibiotics-09-00253-t001:** A comparison of 2014–2016 data to 2016–2019 data, excluding reflective questions and questions 10 and 15. RCGP: Royal College of General Practitioners, UTIs: urinary tract infections.

Number	Question	November 2014–June 2016	July 2016–September 2019	Percentage Change
1	Do you use national or local antibiotic guidance when considering how to treat common infections?	98%	98%	No change
2	Do you analyse and discuss antibiotic prescribing at your practice in comparison to local indicators at least once a year?	67%	73%	6% increase
3	Do you use patient-focused strategies to highlight the importance of responsible antibiotic use? For example, videos and posters in clinical and waiting areas.	71%	94%	23% increase
5	Have you been involved in a practice antibiotic audit in the last two years?	45%	98%	53% increase
6	Do you usually record clinical indications for prescribed antibiotics in patient notes using Read codes or Snomed codes?	75%	80%	5% increase
8	Is the latest antibiotic guidance made available to all temporary prescribers working in your surgery?	63%	55%	8% decrease
9	Do you use back-up/delayed prescribing?	94%	99%	5% increase
12	Do you keep a written record and surgery action plan resulting from antibiotic audits?	62%	81%	19% increase
13	Do you have a strategy to avoid patients reconsulting with other clinicians to obtain antibiotics?	33%	45%	12% increase
14	Have you undertaken any antibiotic-related prescribing clinician courses, for example the TARGET webinars, ‘managing acute respiratory tract infections’ and ‘managing UTIs elearning courses on the RCGP online learning environment within the past two years?	29%	37%	8% increase

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
