# Peer review of "Self-Reported Antimicrobial Stewardship Practices in Primary Care Using the TARGET Antibiotics Self-Assessment Tool"

_antibiotics, 2020, doi:10.3390/antibiotics9050253_

Round 1

Reviewer 1 Report

In this article entitled "Self-Reported Antimicrobial Stewardship Practices in Primary Care Using the TARGET Antibiotics Self-Assessment Tool" Jones et al.  analyzed responses from June 2016 to September 2019 of the SAT available in the TARGET antibiotics toolkit. This article aims to give an update and compared the results with the previous SAT data. In my opinion, this article is significantly sounded and only needs minor revisions (text editing) prior to publication. My only concern is in regard to the Result section (Section 2). From a reader's perspective, this section is hard to follow, especially in section 2.1. Also, I think it will be interesting for the readers if the authors added more of the answer/comments in lines 100 and 153. 

Reviewer 2 Report

            Thank you for the opportunity to review the manuscript, “Self-reported antimicrobial stewardship practices in primary care using the TARGET antibiotics self-assessment tool,” by Leah Ffion Jones, et al. The authors describe data collected from a self-administered survey of antimicrobial stewardship practices performed by 2,373 primary care providers in the United Kingdom. The survey was used between 2016 and 2019 and the authors showed change over time in these practices. In particular, they found that providers required improvement in the following areas: practices to discourage patients from reconsulting to obtain antibiotics, themselves undertaking antibiotic-related training, providing guidance to temporary prescribers, and providing leaflets on stewardship topics to patients in their practices.  Antimicrobial stewardship in the outpatient setting is critical to the control of antimicrobial overuse in medicine generally, and monitoring of practices as described in this manuscript provides an important view into how stewardship practices may be improved.

            The primary weakness of the study is that the survey, because it was a voluntary exercise, very likely reflected some bias in the participants, but it would be difficult to predict whether those who participated were more or less likely to be active in stewardship activities.  The authors recognize the potential for this bias. Otherwise, the study provides interesting insights into the practice of GPs and other primary care providers related to antimicrobial stewardship. The results of this survey, however, are not likely to be relevant in other settings in the world.

            I have the following specific comments and questions for the authors:

  1. It would be useful to report the demographic characteristics of respondents to the SAT to determine if they are representative (geographically, by age, and/or years of experience, etc.) of the larger population of health care workers in similar practices in the UK then the results may be more convincing. Can the authors provide such an analysis?

  1. The Results section, I believe, can be shortened by at least 25%.

  1. Figure 1 perhaps can be included as an on-line appendix; in my opinion the text adequately demonstrates the content of the survey questions.

  1. The quotations provided in the Results (with Figures 2 and 3), in my opinion, add little to the data provided as it is unclear how representative the expressed sentiments are.

  1. In the Discussion, can the authors provide some insight into any implications that their findings may have for stewardship efforts outside of the UK?

Reviewer 3 Report

The work presented is interesting and it was a pleasure reading through it. However, minor corrections are needed and can be addressed through proofreading. For example,

Line 163: Delete “below”

Line 267: Please define NICE

My major concern is that the database and approach used in this study is unique to the UK. It would be good to make suggestions for antimicrobial stewardship in other places. Such organised databases do not exist in most least and middle-income countries for example. What will then guide best practice in such a place? Maybe if access to the data was made publicly available, as was the case initially, then others may learn from the UK experience.

Round 2

Reviewer 2 Report

The authors have responded well to my comments and questions. I have no further comments or questions on the revised version.